# Estimating Heating Load in Residential Buildings Using Multi-Verse Optimizer, Self-Organizing Self-Adaptive, and Vortex Search Neural-Evolutionary Techniques

**Fatemeh Nejati** [1] , **Nayer Tahoori** [2], **Mohammad Amin Sharifian** [3] , **Alireza Ghafari** [4,*] **and Moncef L. Nehdi** [5,*]

1    Department of Art and Architecture, Faculty of Architecture, Khatam University, Tehran 1417466191, Iran
2    Department of Art, Science and Research Branch, Islamic Azad University, Tehran 1477893855, Iran
3    Department of Architecture, Science and Research Branch, Islamic Azad University, Tehran 1477893855, Iran
4    Department of Architecture, Rafsanjan Branch, Islamic Azad University, Rafsanjan 1477893855, Iran
5    Department of Civil Engineering, McMaster University, Hamilton, ON L8S 4M6, Canada
*    Correspondence: alireza.ghafari4@gmail.com (A.G.); nehdim@mcmaster.ca (M.L.N.);
     Tel.: +1-905-525-9140 (ext. 23824) (M.L.N.)

**Abstract:** Using ANN algorithms to address optimization problems has substantially benefited recent research. This study assessed the heating load (HL) of residential buildings' heating, ventilating, and air conditioning (HVAC) systems. Multi-layer perceptron (MLP) neural network is utilized in association with the MVO (multi-verse optimizer), VSA (vortex search algorithm), and SOSA (self-organizing self-adaptive) algorithms to solve the computational challenges compounded by the model's complexity. In a dataset that includes independent factors like overall height and glazing area, orientation, wall area, compactness, and the distribution of glazing area, HL is a goal factor. It was revealed that metaheuristic ensembles based on the MVOMLP and VSAMLP metaheuristics had a solid ability to recognize non-linear relationships between these variables. In terms of performance, the MVO-MLP model was considered superior to the VSA-MLP and SOSA-MLP models.

**Keywords:** self-organizing self-adaptive; vortex search algorithm; multi-verse optimizer; heating load; residential





## 1. Introduction

The heating, ventilation, and air conditioning (HVAC) systems of a freshly constructed building regulate indoor air quality [1]. On the other hand, the rising trend of individuals living in energy-efficient buildings necessitates a thorough comprehension of the entire thermal loads necessary to choose appropriate HVAC systems. Several mathematical and analytic techniques [2–4] have optimized HVAC systems. According to a recent study, machine learning techniques (i.e., inverse modeling) can be used to predict and evaluate the buildings' energy performance [5]. Due to developments in programming sciences and computation, various innovative approaches have been created over the past several years [6–8]. The main goal of these simulations is to make simulations of actual events more practical [8–10]. Using a range of methods (e.g., numerical [11,12], experimental [13,14], empirical [15,16]), scientists have been able to select the most suitable technique for the unsolved problem. Several more conventional processes could be supplanted by machine learning, which has shown promising outcomes. Using various machine-learning programs, it is feasible to solve intricate problems with high accuracy.

The artificial neural network (ANN) [17,18] is a powerful processor capable of simulating a variety of scientific objectives and tasks [19–24]. Due to its neural processors and several layers, the multi-layer perceptron (MLP) [25] is a characteristic form of ANN. The utilization of these processors in simulations involving energy has been effective [26–28]. Using an MLP, researchers can study the relationship between a dependent parameter and

other independent factors. Each dependent parameter is assigned a weight in the neurons of the MLP, which are its processors. The resultant value will then be used to activate a function by combining it with a bias term. The subsequent development of neurons uses this strategy to have a unique mathematically forward progress [29].

Consequently, the MLP has become a "feed-forward instrument" [30]. Analytical approaches were congruent with Ren et al.'s hypothesized ANN results heat loss prediction [31–33]. This model surpasses all others in calculating the strain in a concrete beam's tie section, as Mohammadhassani et al. [34]. Sadeghi et al. [35] utilized an MLP to predict a residential structure's cooling and heating demands. A sensitivity analysis also indicated the ideal network response. Sholahudin and Han [36] employed the Taguchi method to develop a simplified dynamic ANN to accurately predict heat loss (HL) in an HVAC system. Several prior studies [37–39] have proved the efficacy of ANNs in energy modeling. In addressing energy-related issues, fuzzy networks [40], random forests, and support vector machines have all been useful [41–43].

In energy analysis, metaheuristic scholars have been increasingly interested in HVAC systems and energy analysis [44–49]. Martin et al. [50] calibrated the HVAC subsystem component via a metaheuristic and sensitivity analysis. Bamdad Masouleh [51] implemented ant colony optimization to optimize energy. Moreover, several benchmarks revealed that the proposed models excelled in traditional methods. Numerous research studies have demonstrated that machine learning models can benefit from various techniques [52–55]. As part of their research, Zhou et al. [56] investigated how to best estimate the HL and CL by ANN, utilizing ABC and PSO applied to an ANN [57]. The PSO outperformed the other algorithm by approximately 22 to 24 percent, demonstrating that both approaches are effective. Bui et al. [58] used a firefly technique based on electromagnetism to optimize the ANN for calculating energy use. Researchers discovered that hybrid approaches were more exact than a conventional ANN technique. In this sense, Moayedi et al. [59] assessed the performance of grasshopper optimization algorithm (GOA) and grey wolf optimization (GWO) optimizers in conjunction with an ANN, for estimating the heating load of green residential construction [60]. As a result of these tactics, the prediction error dropped from 2.9859 to 2.4460 and 2.2898, respectively.

Metaheuristic approaches have developed to overcome common computing restrictions, including local minima [61–70]. Employing these methods to find the intelligent models' training would result in very accurate predicting models for various goals [71,72]. Because there are so many optimization methods, comparative research on the next generation of metaheuristics is necessary.

Environmentally and economically, finding a realistic model for thermal load modeling could be advantageous. The main goal of this article is to forecast the heating and cooling load via metaheuristic algorithms and check whether these algorithms can predict the heating and cooling load precisely. Metaheuristic optimizers, such as the multi-verse optimizer (MVO), self-organizing and self-adaptive (SOSA), and vortex search algorithm (VSA), are being evaluated to discover whether they can aid in estimating the HL. Also, these three methods are compared, and the best one is presented at the end of the task.

## 2. Established Database

The connection between these influencing factors and parameters must be investigated to estimate a parameter. Hence, the supplied data must be accurate. A total of 768 thermal load scenarios are employed to train, test, and validate the models in this work. The data was initially developed by Tsanas and Xifara, who analyzed the heating load and cooling load of various residential buildings [73]. Due to their work, a valuable dataset was compiled and made accessible for download at https://archive.ics.uci.edu/ml/datasets/Energy+efficiency (accessed on 15 July 2022). Overall height (OH), roof area (RA), glazing area (GA), wall area (WA), relative compactness (RC), orientation (OR), surface area (SA), and glazing area distribution (GAD) are independent factors identified to affect the HL

output parameters. A box plot of the heating load and input components is displayed in Figure 1.

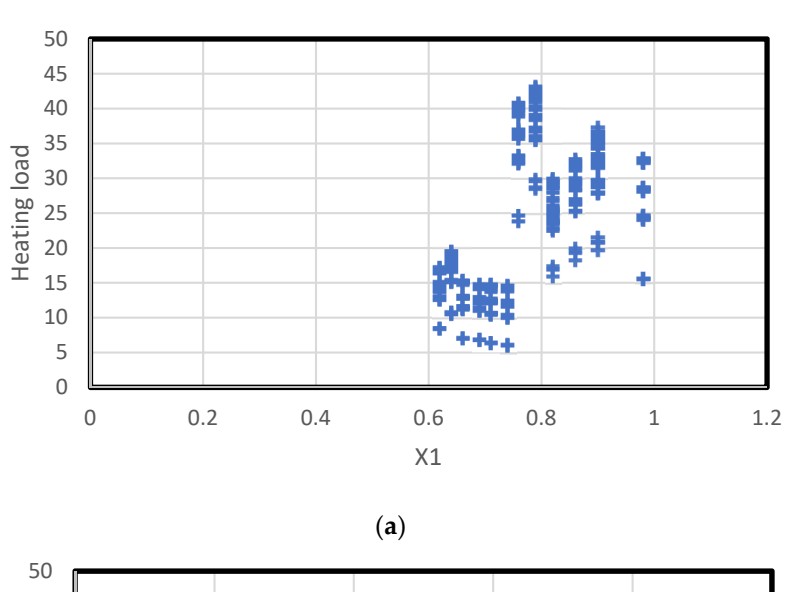

(**a**)

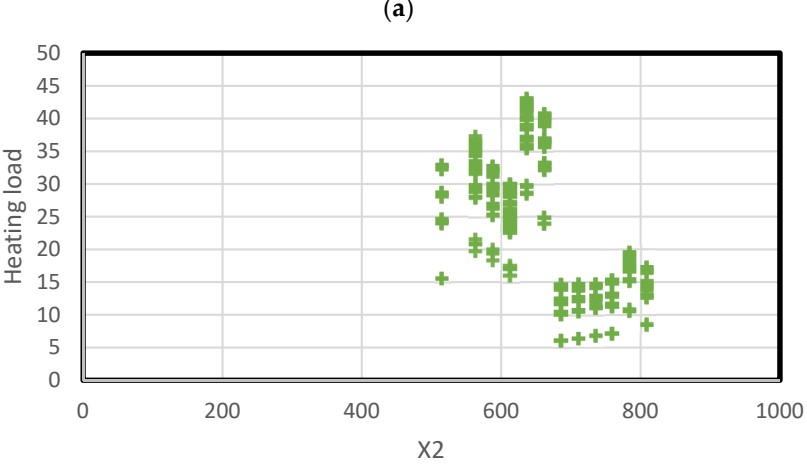

(**b**)

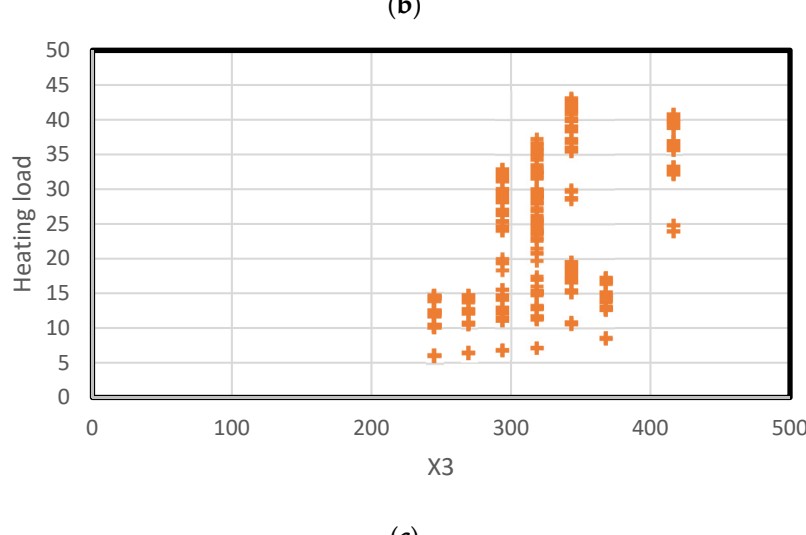

(**c**)

**Figure 1.** *Cont.*

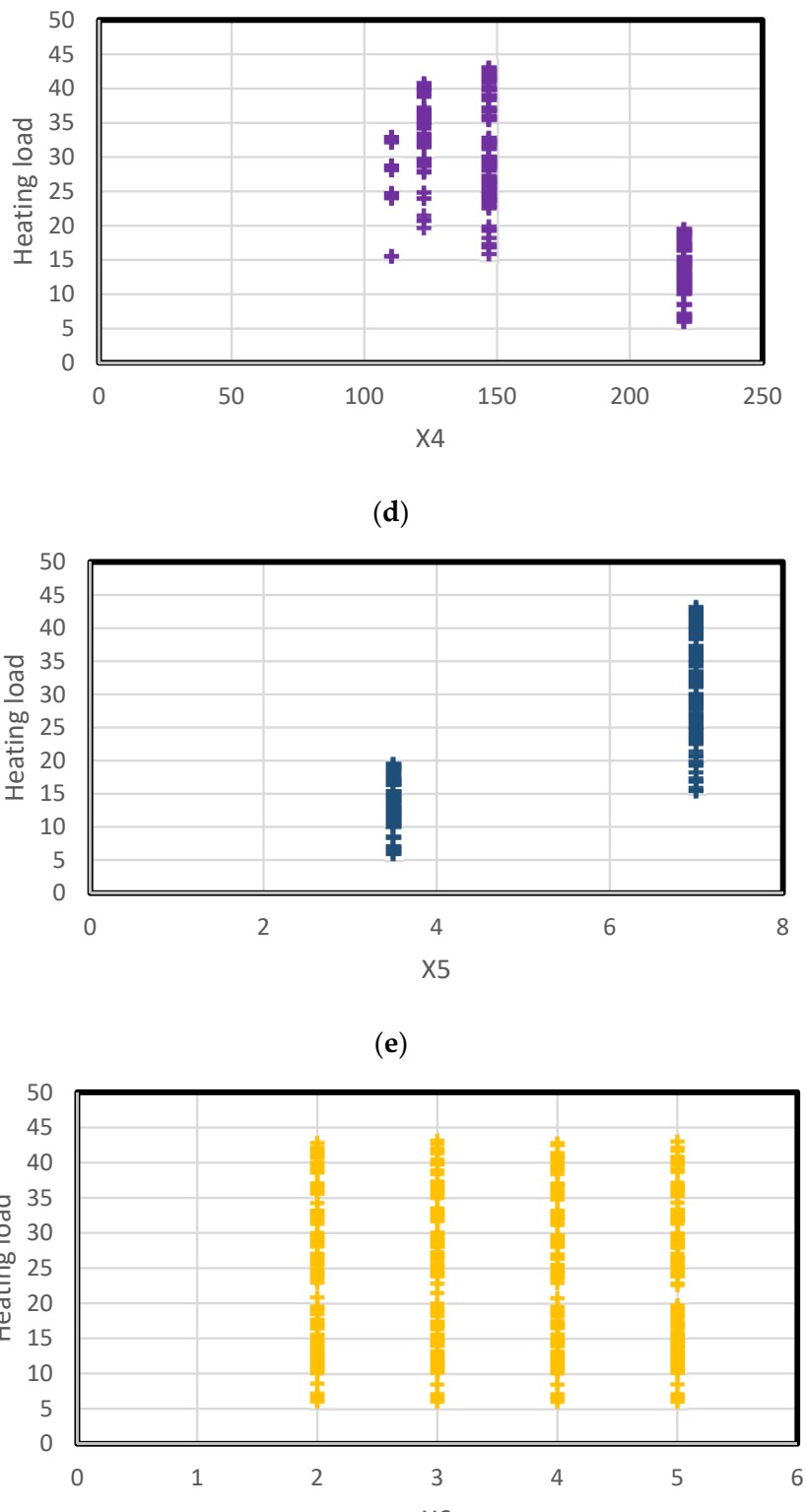

(**d**)

(**e**)

(**f**)

**Figure 1.** *Cont.*

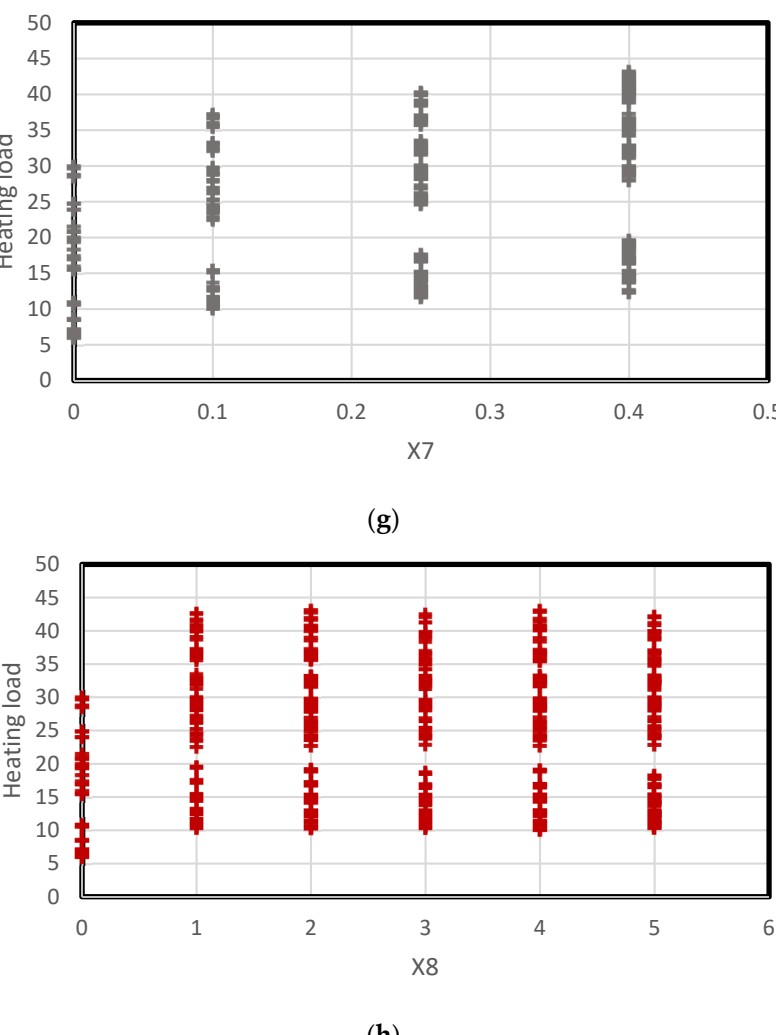

(**g**)

(**h**)

**Figure 1.** Box plot of used dataset variations with the heating load. (**a**) Relative compactness (RC), (**b**) surface area, (**c**) wall area, (**d**) roof area, (**e**) overall height, (**f**) orientation, (**g**) glazing area, (**h**) glazing area distribution, with the heating load.

## 3. Methodology

This research examines an ANN with three novel optimizers, MVO, SOSA, and VSA, to test their investigation of how they affect the limits of a typical neural network. These algorithms seek better hyperparameters than those proposed by more conventional learning methods (backpropagation and Levenberg–Marquardt).

### 3.1. Multilayer Perceptron

Multilayer perceptrons, a type of neural network, have recently been demonstrated to be a viable alternative to conventional statistical methods [74]. Hornik et al. (1989) [75] demonstrated that the MLP could simulate any smooth and measurable function. Despite other methods, the MLP method does not consider data processing. This method can model and teach complex nonlinear functions to generalize correctly using previously unexplored new data. These properties make it a possible alternative to statistical and numerical modeling techniques. The multilayer perceptron has several atmospheric scientific uses, as will be demonstrated.

Figure 2 depicts the predefined connection between the main inputs and output(s) vectors for the multilayer perceptron, a network of fundamentally interconnected neurons or nodes. Each network node's output signals and weights are derived from a primary activation function or nonlinear transfer. The MLP can only model linear functions if the transfer

function is linear. The node's output can serve as an input for other network-connected nodes for each network-connected node. In light of this, the multilayer perceptron is a feed-forward neural network. There are a variety of structural configurations for multilayer perceptrons, but they all contain layers of neurons. The input layer serves as a conduit for data transfer from the input layer to other network layers. A multilayer perceptron's input and output vectors can be expressed as single vectors (Figure 2). An MLP structure consists of multiple hidden layers and one output layer. Multilayer perceptron refers to a network in which each node is interconnected in the layers above and below with every other node.

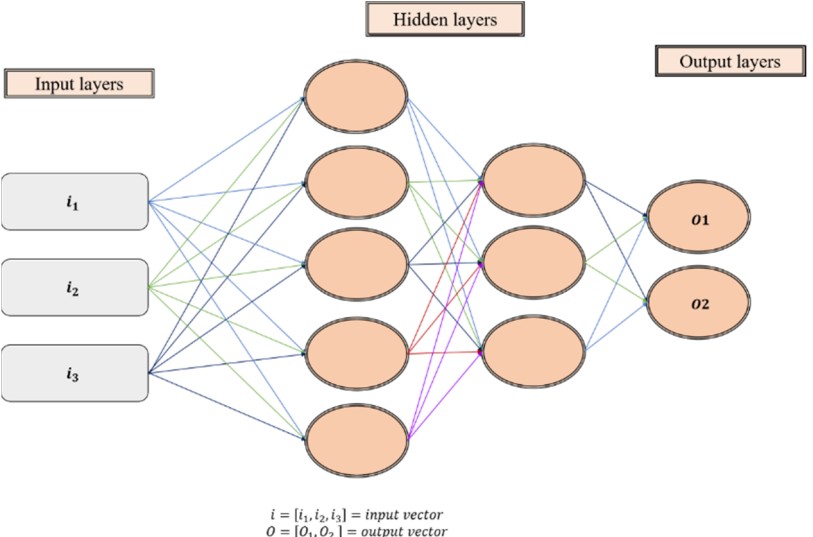

$$i = [i_1, i_2, i_3] = input\ vector$$
$$O = [O_1, O_2] = output\ vector$$

**Figure 2.** A two hidden layers multilayer perceptron.

As proven, multilayer perceptrons can estimate any computable function between two sets of input and output vectors by selecting a fine collection of linking weights and transfer functions [75]. A multilayer perceptron is capable of learning new abilities by training. You will require input and output vector-based training data to learn a new algorithm. A multilayer perceptron decides on the network's weights until the required input-output mapping is reached. It can only acquire knowledge in the presence of an observer. When training an MLP, it is possible that its output for a given input vector may not match the anticipated output. The difference between the actual and desired outputs characterizes error signals. Adjusting the direct networks depending on this error signal during training can help lower the total error of the MLP. A multilayer perceptron can be trained in various methods with several different algorithms. Once trained with adequate training data, the multilayer perceptron can generalize to new, unknown inputs.

### 3.2. Multi-Verse Optimizer (MVO)

The multi-verse optimizer [76] is known as a growing metaheuristic algorithm that tries to mimic the laws of a multi-verse theory. It is a relatively recent development. Parallel universe theories, including the presence of black, white, and wormholes, were the primary source of inspiration for the design of this optimizer. A population-based stochastic method is employed to determine the global optimum for optimization problems [77]. To update the answers using this method, the probability of wormhole existence (*WEP*) and the rate of travel (*TDR*) must first be computed. These parameters determine the frequency and magnitude of solution changes during the optimization process and are formulated as:

$$WEP = a + t \times \left( \frac{b-a}{T} \right) \tag{1}$$

The total iterations' number is $T$, corresponding to the minimum, $b$ to the maximum, and $t$ to the current iteration.

$$TDR = 1 - \frac{t^{1/P}}{T^{1/P}} \tag{2}$$

$p$ indicates the exploitation accuracy. $P$ is the most essential *TDR* measure. The emphasis on exploitation increases as the value of this choice rises.

The following equation can be used to update the solution positions when *WEP* and *TDR* have been calculated:

$$x_i^j \begin{cases} \begin{cases} x_j + TDR + \left( (ub_j - lb_j) * r_4 + lb_j \right) & if \ r_3 < 0.5 \\ x_j - TDR + \left( (ub_j - lb_j) * r_4 + lb_j \right) & if \ r_3 \geq 0.5 \end{cases} & if \ r_2 < WEP \\ x_{roulette\ Wheel}^j & if \ r_2 \geq WEP \end{cases} \tag{3}$$

where $x_j$ is set to be the $j$th element from the best predefined individual, *WEP*, *TDR* are coefficients, $lb_i$ and $ub_i$ are the lower and upper bounds of the $j$th element, $r_2$; $r_3$; $r_4$ are randomly generated numbers drawn from the interval of [0, 1], $x_i^j$ represents the $j$th parameter in $i$th individual, and $x_{roulette\ Wheel}^j$ does the roulette wheel selection mechanism to pick the $j$th element of a solution.

This equation can be used to compute a new solution position and compare it to the most recent best-in-class participant in the *WEP*. If $r_3$, a random number in the interval [0, 1], is less than 0.5, then an optimal solution value for the $j$th dimension requires a solution. By increasing *WEP* during optimization, MVO increases the use of the most proper solution so far.

### 3.3. Self-Organizing and Self-Adaptive (SOSA)

Self-organization (SO) parallels the biologically inspired notions of emergence and swarm intelligence very closely. Frequently, in this technique, SO and emergence are conflated. De and Holvoet (2005) [78] examine the phrase's origins and the difference between the two conceptions. This is known as SO:

SO is an adaptive and dynamic computational process through which systems retain their structure independently of external stimuli [78,79]. However, SO can also refer to the emergence-causing process [80,81]. In addition, ref. [82] differentiates between the terms called strong SO schemes with no explicit central internal or external control and weak SO systems with some central internal control. SO and emergent systems are separate concepts, although they share one characteristic: the absence of direct exterior control. Although the external effect on self-organized systems is studied more thoroughly in directed SO, less attention has been paid to it in the context of unguided SO [83]. In this text, external effect is characterized as either specific or non-specific, with specific influence suggesting straight control on the functional structure or temporal, spatial, or other non-specific impacts indicating that the system determines its response to an external stimulus. Consequently, Prokopenko (2009) [83] defines SO guidance as the potential limiting of the domain or extent of functions/structures, or selecting a subset of the multiple alternatives that the dynamics might take.

According to ref. [78], the main distinction between SO and emergence is that individual entities are informed of the systems planned by global behavior in the former scenario. Consequently, self-organization may be considered a weak kind of emergence. Utilizing feedback loops is a common and straightforward method for achieving SO. Components of the system monitor the state, interpret it according to the expected behavior, and initiate the required actions. This method is also employed by "single entity systems." This notion is referred to as self-adaptation [84,85]. Self-adaptation happens when a decentralized system composed of several entities adapts to external changes. Self-adaptation within the context of software is set as follows: SA software modifies its behavior in response to modifications within its operating environment. The operating environment refers to everything the software system may see, including human input, sensors and external hardware devices, and programmed instrumentation [86].

### 3.4. Vortex Search Algorithm (VSA)

Ölmez and Doğan [87] initially developed the vortex pattern generated by the vertical flow of stirred fluids to design the VSA algorithm. As with countless other methods, the algorithm seeks to balance exploratory and exploitative actions. The VSA uses an adaptive step-size-adjustment method to determine the optimal response. Consequently, exploratory behavior is accounted for in the early phases of the VSA, resulting in a better global search capability. In the following, the optimal response is achieved by employing an exploitative strategy around the suggested replies [88].

The vortex is depicted by stacked circles, assuming a set in two dimensions. Given $U$ and $L$ as the current space's boundaries, Equation (4) produces the starting point $\lambda_0$. of the outer circle:

$$\lambda_0 = \frac{U + L}{2} \tag{4}$$

Then, several neighbor solutions $Ct(s)$ are generated at random. This production makes use of a Gaussian distribution technique.

$$C_0(s) = \{S_1, S_2, \ldots, S_g\} \qquad g = 1, 2, \ldots, z, \tag{5}$$

where $t$ is the number of cycles and $z$ represents the total number of potential solutions. Let $x$ and $\Sigma$ be the vector and covariance matrix of the random variable. The multivariate Gaussian distribution is denoted by Equation (6):

$$P(x|\lambda, \Sigma) = \frac{1}{\sqrt{(2\pi)^D \Sigma}} \exp\left\{\frac{-1}{2}(x - \lambda)^T \Sigma (x - \lambda)\right\}, \tag{6}$$

where $D$ is the magnitude of the issue and is the mean vector introduced as sample.

The main distribution will be spherical if the off-diagonal elements are uncorrelated and the co-variance matrix values have similar variances (circular for two-dimensional concerns). $I$, where $I$ is a $D \times D$ identity matrix and $\sigma^2$ is the distribution's variance, $\Sigma$ may be written as follows:

$$\Sigma = \sigma^2 \times [I]_{D \times D}. \tag{7}$$

Using Equation (8), the initial standard deviation of the distribution is computed ($\sigma_0$). This parameter may correspond to $r_0$. (which requires significant values) [89]:

$$\sigma_0 = \frac{\text{maximum } (U) - \text{minimum } (L)}{2} \tag{8}$$

As is well known, the essential concept of metaheuristic algorithms for enhancing the final result is to update the obtained answers. During the VSA selection phase, the current $\lambda_0$ is replaced with the most promising alternative. This requires the proposed solution to exist inside the given space. This item is assessed using the Equation (9).

$$\begin{cases} s_g^i = rand \cdot (U^i - L^i) + L^i, & if \ s_g^i < L^i \\ s_g^i = rand \cdot (U^i - L^i) + L^i, & if \ s_g^i > U^i \end{cases} \tag{9}$$

where rand is a random integer with uniform distribution.

The best answer discovered thus far is then applied to the second (or inner) circle's center. After successively decreasing the effective radius of the current solution, a new group of solutions (C1(s)) is produced close to it. Repetitioning the same approach might yield a more viable answer [89]. Other researches have also described the VSA well [90,91].

### 4. Results and Discussion

This study analyzes the HL approximation capabilities of three unique neural network upgrades described in Section 1. The algorithms are synthesized using an MLP neural

network to accomplish this objective. Each approach uses a unique search strategy to get the optimal computational weights for the MLP (and biases).

As is commonly known, the size and number of neurons contained inside a hidden layer define the MLP's structure. Therefore, these parameters must initially be modified. Numerous studies have demonstrated that a single hidden layer is excellent at simulating complicated processes [92,93]. However, the hidden neurons' optimal number was established by trial and error. Among the designs studied, $8 \times 6 \times 1$ demonstrated the most promising performance (where the middling layer contained 1, 2, 3, ... , 10 neurons). Figure 2 illustrates the used MLP.

### 4.1. Accuracy Indicators

Mean absolute error (MAE) as the first used statistical index and root mean square error (RMSE) as the second index was specified for assessing the potential errors in proposed structures. Equations (10) and (11) produce are used for RMSE and MAE. Additionally, Equation (12) defines the coefficient of determination ($R^2$) required to compute the compatibility between the measured and predicted HLs:

$$\text{MAE} = \frac{1}{U} \sum_{i=1}^{U} \left| S_{i_{observed}} - S_{i_{predicted}} \right| \tag{10}$$

$$\text{RMSE} = \sqrt{\frac{1}{U} \sum_{i=1}^{U} \left[ (S_{i_{observed}} - S_{i_{predicted}}) \right]^2} \tag{11}$$

$$R^2 = 1 - \frac{\sum\limits_{i=1}^{U} \left( S_{i_{predicted}} - S_{i_{observed}} \right)^2}{\sum\limits_{i=1}^{U} \left( S_{i_{observed}} - \overline{S}_{observed} \right)^2} \tag{12}$$

$S_{i_{observed}}$ and $S_{i_{anticipate}}$ represent the measured and expected HLs, respectively, in these equations. In addition, $U$ represents the number of recordings, whereas $S_{observed}$ is the average of the observed HLs.

### 4.2. Combining the MLP with Hybrid Optimizers

After combining hybrid algorithms with the MLP, three ensembles of MVO-MLP, SOSA-MLP, and SOSA-MLP are constructed. Each costume is supplied with training data to determine the relationship between associated parameters and heating load. One thousand repetitions are assessed for each model's optimization behavior in order to carry out the optimization. The objective function is represented using the RMSE of each iteration's findings. In swarm-based optimization algorithms, the population size is a critical variable. Ten distinct population sizes (50, 100, 150, 200, 250, 300, 350, 400, 450, and 500) are evaluated for each proposed model, and the population size results in the lowest MSE chosen as the optimal population size. The MSEs for all calculated iterations are shown in Figure 3. The populations with the lowest RMSE values (0.3540, 8.8064, and 0.2887, respectively) are 300, 4500, and 100 for MVO-MLP, SOSA-MLP, and VSA-MLP, respectively. The SOSA-MLP method, on the other hand, is less sensitive than the other two; the explanation for this may be found in the optimization approaches' characteristics. Figure 4 also displays the RMSE values achieved for different levels of complexity over all rounds.

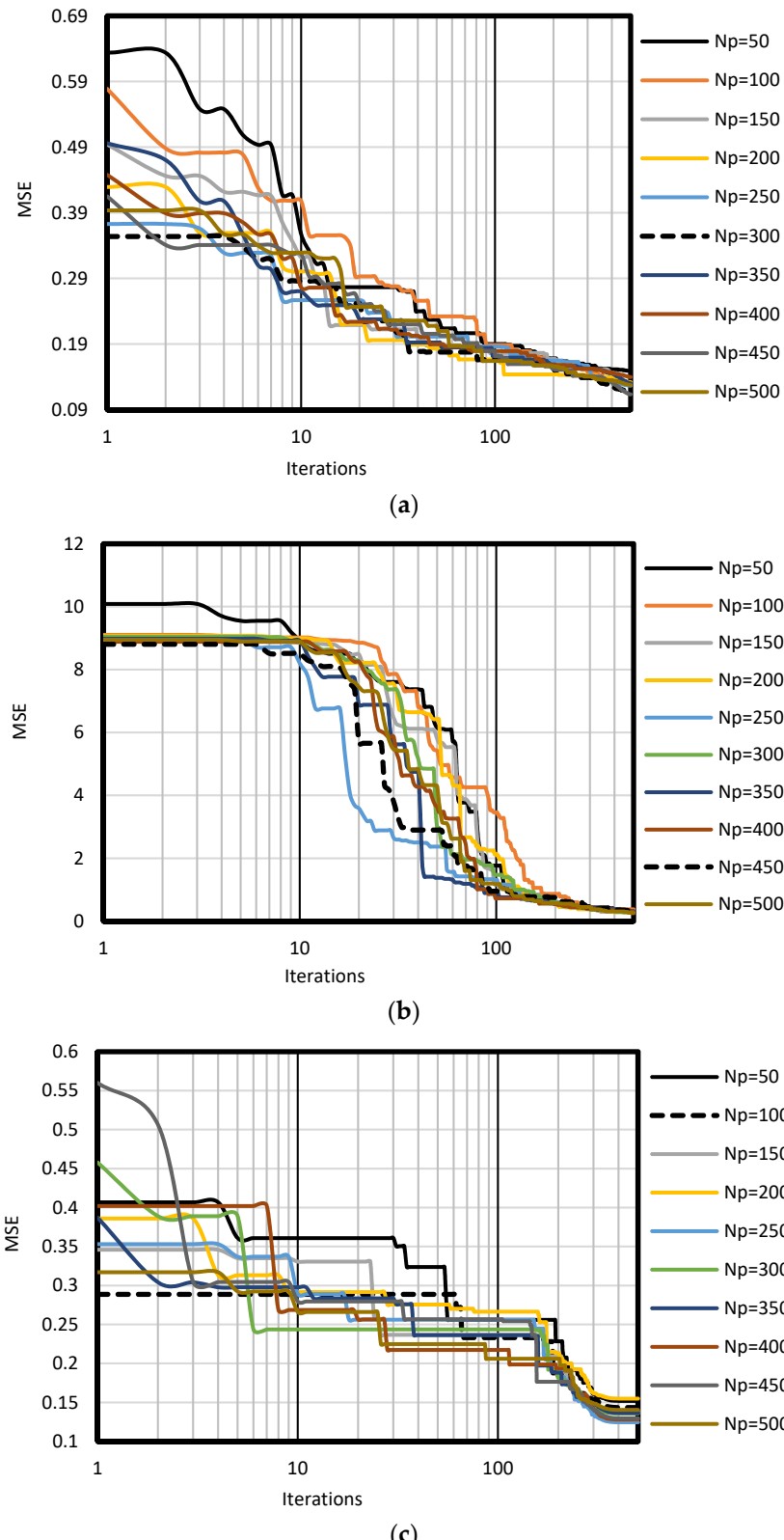

**Figure 3.** Model iterations versus the variation of MSE; (**a**) MVO-MLP, (**b**) SOSA-MLP, (**c**) VSA-MLP.

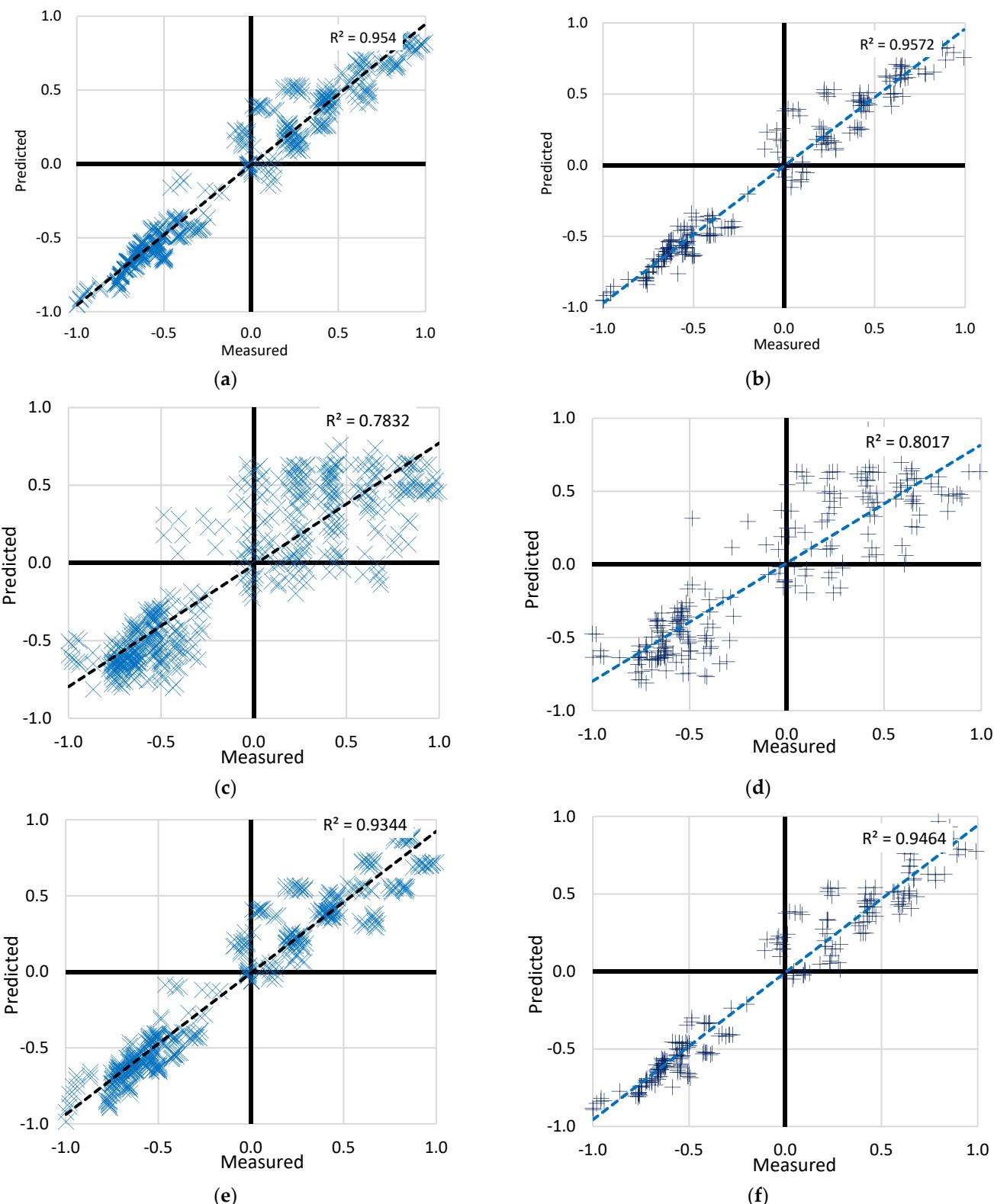

**Figure 4.** The accuracy of the best-fit proposed model for the (**a**) MVO-MLP training dataset, (**b**) MVO-MLP testing dataset, (**c**) SOSA-MLP training dataset, (**d**) SOSA-MLP testing dataset, (**e**) VSA-MLP training dataset, and (**f**) VSA-MLP testing dataset.

The value of $R^2$ for three methods of MVO, SOSA, and VSA is (0.977 and 0.978), (0.885 and 0.895), and (0.974 and 0.975) for testing and training phases, respectively. Also, in the case of RMSE, MVO, SOSA, and VSA have the value of (0.117 and 0.110), (0.255 and

0.239), and (0.124 and 0.112) in the training and testing phases, respectively. These results show that the lowest value of RMSE and the highest value of $R^2$ are related to the MVO technique, indicating the best performance of MVO-MLP. According to $R^2$ and RMSE values (Tables 1–4), the second technique for predicting HL and CL is VSA-MLP, and the last is SOSA-MLP.

**Table 1.** The network results for the MVO-MLP.

| Population Size | Network Result | | | | Scoring | | | | Total Score | RANK |
|---|---|---|---|---|---|---|---|---|---|---|
| | Train | | Test | | Train | | Test | | | |
| | $R^2$ | RMSE | $R^2$ | RMSE | $R^2$ | RMSE | $R^2$ | RMSE | | |
| 50 | 0.962 | 0.149 | 0.964 | 0.143 | 1 | 1 | 1 | 1 | 4 | 10 |
| 100 | 0.972 | 0.130 | 0.974 | 0.120 | 5 | 5 | 4 | 5 | 19 | 6 |
| 150 | 0.972 | 0.129 | 0.975 | 0.117 | 6 | 6 | 6 | 7 | 25 | 5 |
| 200 | 0.971 | 0.132 | 0.975 | 0.119 | 3 | 3 | 5 | 6 | 17 | 7 |
| 250 | 0.973 | 0.127 | 0.976 | 0.115 | 8 | 8 | 7 | 8 | 31 | 3 |
| 300 | 0.977 | 0.117 | 0.978 | 0.110 | 9 | 9 | 9 | 10 | 37 | 1 |
| 350 | 0.971 | 0.130 | 0.973 | 0.123 | 4 | 4 | 3 | 4 | 15 | 8 |
| 400 | 0.967 | 0.140 | 0.966 | 0.137 | 2 | 2 | 2 | 2 | 8 | 9 |
| 450 | 0.978 | 0.113 | 0.980 | 0.127 | 10 | 10 | 10 | 3 | 33 | 2 |
| 500 | 0.973 | 0.127 | 0.976 | 0.115 | 7 | 7 | 8 | 9 | 31 | 3 |

**Table 2.** The network results for the SOSA-MLP.

| Population Size | Network Result | | | | Scoring | | | | Total Score | RANK |
|---|---|---|---|---|---|---|---|---|---|---|
| | Train | | Test | | Train | | Test | | | |
| | $R^2$ | RMSE | $R^2$ | RMSE | $R^2$ | RMSE | $R^2$ | RMSE | | |
| 50 | 0.810 | 0.343 | 0.806 | 0.355 | 2 | 2 | 2 | 2 | 8 | 9 |
| 100 | 0.776 | 0.378 | 0.781 | 0.384 | 1 | 1 | 1 | 1 | 4 | 10 |
| 150 | 0.874 | 0.289 | 0.893 | 0.274 | 6 | 7 | 6 | 7 | 26 | 4 |
| 200 | 0.889 | 0.289 | 0.894 | 0.278 | 10 | 6 | 7 | 6 | 29 | 3 |
| 250 | 0.881 | 0.307 | 0.898 | 0.302 | 7 | 4 | 9 | 5 | 25 | 5 |
| 300 | 0.871 | 0.278 | 0.836 | 0.304 | 4 | 9 | 4 | 4 | 21 | 7 |
| 350 | 0.832 | 0.327 | 0.807 | 0.342 | 3 | 3 | 3 | 3 | 12 | 8 |
| 400 | 0.884 | 0.285 | 0.899 | 0.270 | 8 | 8 | 10 | 8 | 34 | 2 |
| 450 | 0.871 | 0.293 | 0.880 | 0.255 | 5 | 5 | 5 | 9 | 24 | 6 |
| 500 | 0.885 | 0.255 | 0.895 | 0.239 | 9 | 10 | 8 | 10 | 37 | 1 |

**Table 3.** The network results for the VSA-MLP.

| Population Size | Network Result | | | | Scoring | | | | Total Score | RANK |
|---|---|---|---|---|---|---|---|---|---|---|
| | Train | | Test | | Train | | Test | | | |
| | $R^2$ | RMSE | $R^2$ | RMSE | $R^2$ | RMSE | $R^2$ | RMSE | | |
| 50 | 0.961 | 0.152 | 0.965 | 0.140 | 2 | 2 | 2 | 3 | 9 | 9 |
| 100 | 0.965 | 0.143 | 0.967 | 0.135 | 3 | 3 | 3 | 4 | 13 | 8 |
| 150 | 0.968 | 0.138 | 0.968 | 0.133 | 5 | 5 | 4 | 5 | 19 | 6 |
| 200 | 0.959 | 0.155 | 0.964 | 0.141 | 1 | 1 | 1 | 1 | 4 | 10 |
| 250 | 0.974 | 0.124 | 0.977 | 0.112 | 10 | 10 | 10 | 10 | 40 | 1 |
| 300 | 0.969 | 0.136 | 0.973 | 0.122 | 7 | 7 | 8 | 8 | 30 | 3 |
| 350 | 0.968 | 0.136 | 0.970 | 0.128 | 6 | 6 | 6 | 7 | 25 | 4 |
| 400 | 0.972 | 0.128 | 0.974 | 0.120 | 9 | 9 | 9 | 9 | 36 | 2 |
| 450 | 0.972 | 0.130 | 0.973 | 0.140 | 8 | 8 | 7 | 2 | 25 | 4 |
| 500 | 0.967 | 0.140 | 0.970 | 0.129 | 4 | 4 | 5 | 6 | 19 | 6 |

**Table 4.** Selection of the best fit structures among the most accurate items of each model.

| | Swarm Size | Training Dataset | | Testing Dataset | | Scoring | | | | Total Score | Rank |
|---|---|---|---|---|---|---|---|---|---|---|---|
| | | RMSE | R2 | RMSE | R2 | Training | | Testing | | | |
| MVOMLP | 300 | 0.977 | 0.117 | 0.978 | 0.11 | 3 | 3 | 3 | 3 | 12 | 1 |
| SOSAMLP | 500 | 0.885 | 0.255 | 0.895 | 0.239 | 3 | 3 | 1 | 1 | 8 | 2 |
| VSAMLP | 250 | 0.974 | 0.124 | 0.977 | 0.112 | 2 | 2 | 2 | 2 | 8 | 2 |

According to Figure 3, the MVO method has a little more constrained convergence curve than the other methods. This shows that this approach decreases error rates when ANN parameters are altered. As a result, the algorithm's findings are given to develop a prediction model. Referring to Figure 2, the output of the most recent neuron consists of seven parameters (one bias and six weights). This neuron is nourished by six layers of neurons, each responsible for nine parameters (one bias and eight weights). The network consists of 61 optimized variables with metaheuristic methods.

### 4.3. Prediction Results

In this section, the reliability of the applied models is assessed by considering both the outputs (i.e., the predicted HLs) to the target values (i.e., the measured HLs). Figure 5 illustrates the outcomes of the training phase by displaying the difference between each pair of output and HL goals. During this phase, the error rate for the MVO-MLP, SOSA-MLP, and VSA-MLP range between $[-0.000034913$ and $0.11776]$, $[-0.011611$ and $0.25559]$, and $[-5.4249 \times 10^{-5}$ and $0.12416]$, respectively. The preceding section indicates that the RMSE values are 0.3540, 8.8064, and 0.2887. In addition, the estimated MAEs of the three models (0.08499, 0.19662, and 0.088861) demonstrate a small degree of training error. Moreover, the computed R2 values indicate that greater than 93% of the objective and output HLs are consistent.

### 4.4. Efficiency Comparison

The models with the lowest RMSE (or MAE) and the highest $R^2$ are chosen as the most exact HL predictors, considering the learning and prediction stages. Table 4 displays the accuracy standards that must be satisfied to attain this objective. As demonstrated, the MLP constructed utilizing the MVO's weights and biases provide the most accurate knowledge of the HL and predicting it. The VSA appears as the second possible optimizer after the MVO. This study's MVO and VSA algorithms appear to outperform previously proposed models in the training and testing phases. For example, six different MLP network's hybrids (for instance, based on other hybrid techniques, such as whale optimization algorithm (WOA) [94], ABC [95], PSO [96], the salp swarm algorithm (SSA) [97], wind-driven optimization (WDO) [98], the spotted hyena optimization (SHO) [99], the imperialist competitive algorithm (ICA) [100], GOA [101], the genetic algorithm (GA) [102], and GWO [103]) were utilized to estimate the HL by using the same dataset. This suggests that the objective of developing more effective HL assessment tools has been met.

### 4.5. Discussion

In several engineering applications, the superiority of intelligent computational techniques over conventional and even solid experimental methods is well acknowledged. In addition to appropriate accuracy, the simplicity of applying these models is a determining factor in their application. In energy-efficiency studies, for instance, forward modeling methodologies (low capabilities for inhabited buildings [104]) and prevalent simulation software may have limitations (low capabilities for occupied buildings [104]). (Different accuracy of simulation [105]). Consequently, like the models reported in this study, indirect evaluative models outperform destructive and expensive methods. This is emphasized

further when an optimal strategy is created using metaheuristic methods [106]. In other words, these optimization techniques yield competent ensembles that function optimally.

Realistic applications for the offered approaches may be developed in terms of applicability. Here are two illustrations:

The developed technique can provide an accurate estimate of the needed heating thermal load for an upcoming construction project based on the size and features of the structure [26,107,108]. Engineers and property owners might benefit from the models when developing HVAC systems. Another early-stage support for reconstruction projects is modifying structural design and architecture based on input parameters. Consequently, it is also feasible to examine the effect of each input parameter separately to comprehend the thermal load behavior. Although the trend is not predictable nor regular, the MVO-MLP predicts it precisely. Consequently, this approach may yield approximations of real-world structures that are correct.

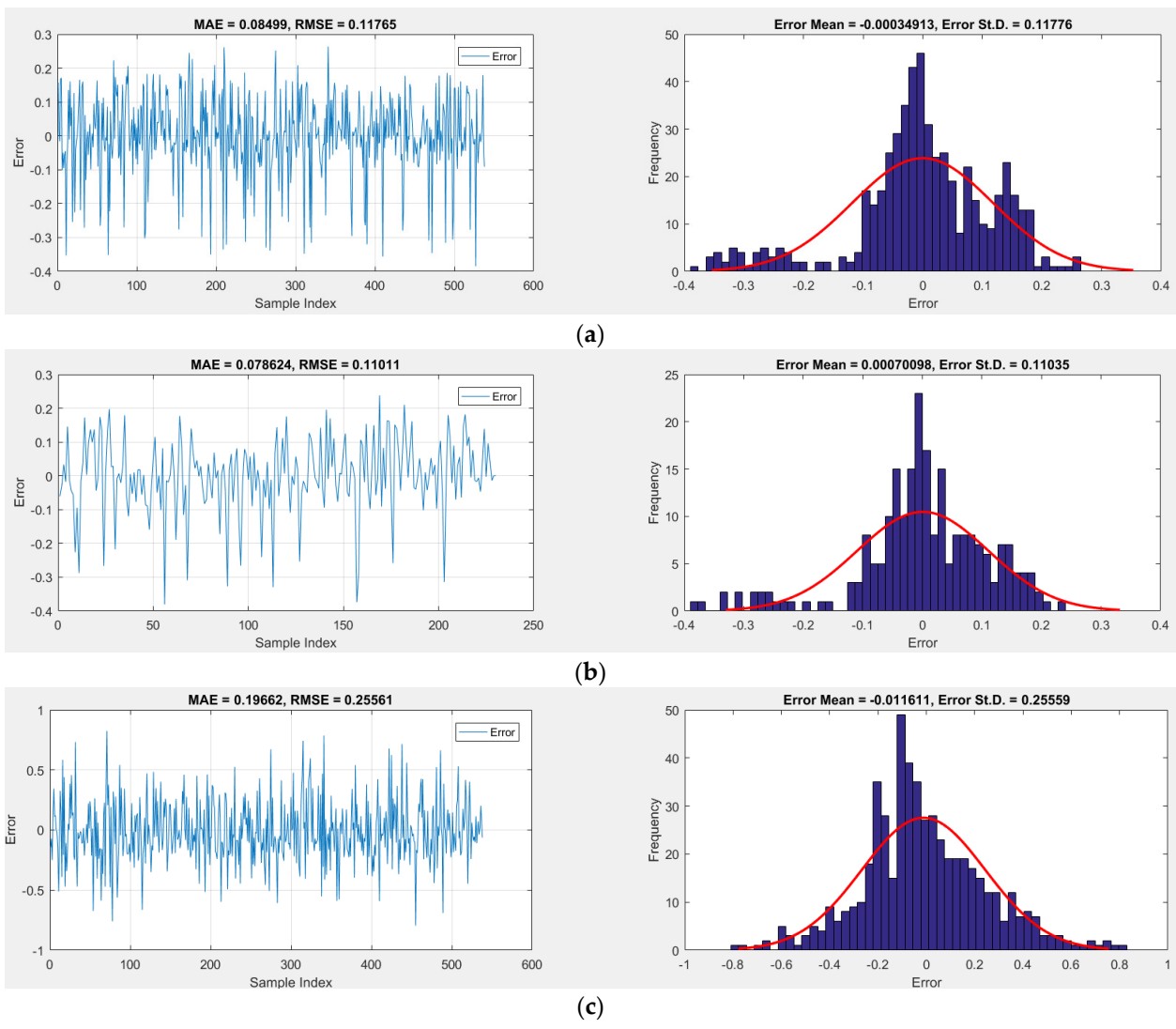

**Figure 5.** *Cont.*

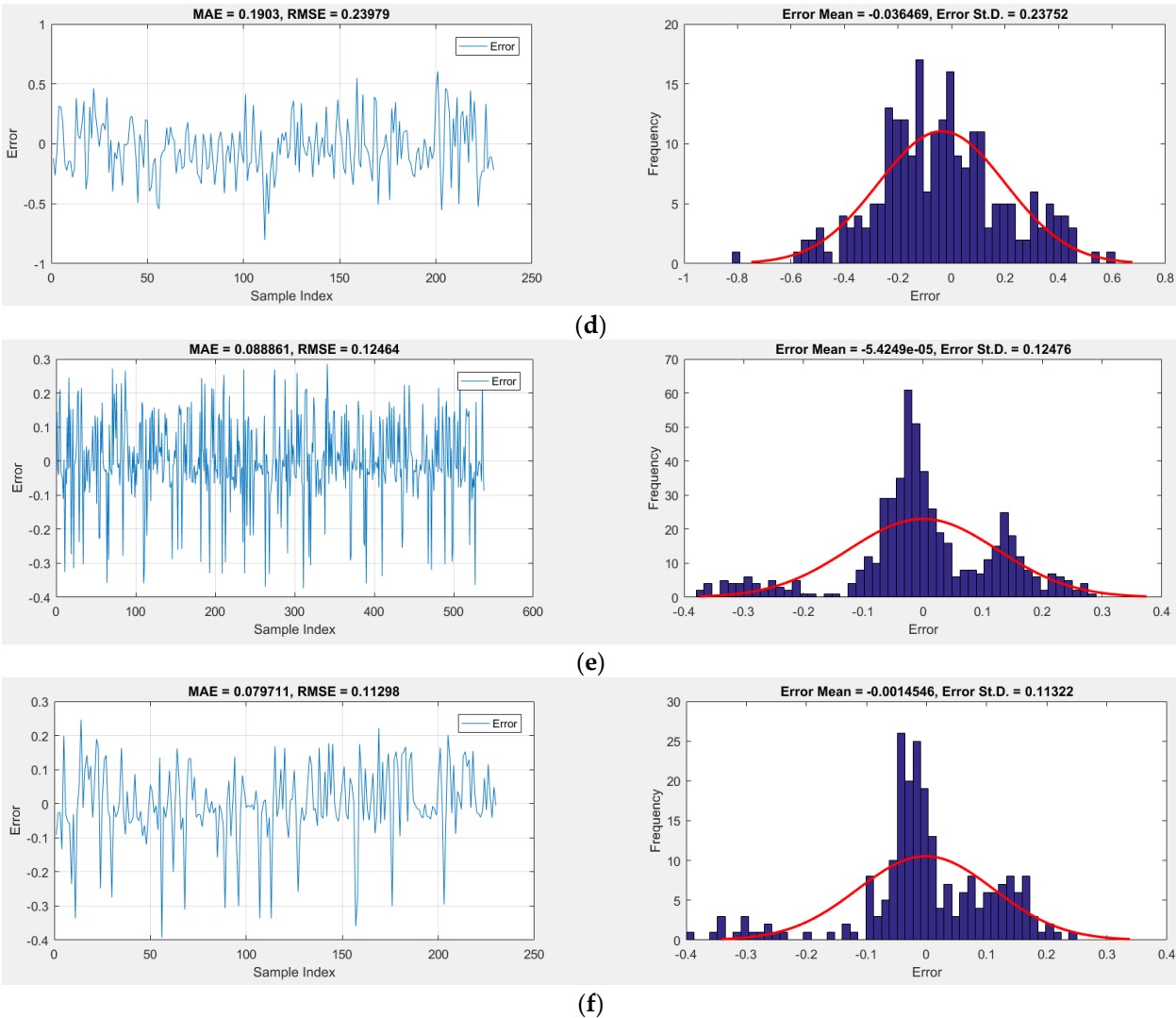

**Figure 5.** The error analysis for the best-fit proposed model for the (**a**) MVO-MLP training dataset, (**b**) MVO-MLP testing dataset, (**c**) SOSA-MLP training dataset, (**d**) SOSA-MLP testing dataset, (**e**) VSA-MLP training dataset, and (**f**) VSA-MLP testing dataset.

Even if there are several benefits to addressing an optimization problem, it is essential to commit the time necessary to discover a global solution. Consequently, achieving a balance between model time economy and precision may impact selecting the most efficient model. Nevertheless, according to the authors, lowering the complexity of the problem space and locating better solutions may be as simple as configuring the hyper-parameters of optimizers correctly and doing feature validity analysis. In contrast, the MVO model was the most precise; this requires establishing the optimal time and accuracy-based method. In projects in which time is not a factor, for instance, it makes sense to choose the most precise technique (regardless of how time-consuming), but in time-sensitive applications, a tolerance for accuracy may be considered in order to find a speedier solution. However, the models' overall performance was comparable, and it should be emphasized that all versions would be adequate for real-world applications. Table 5 indicates the previous research focused on heating load prediction. Noting that the outcomes were less accurate, either using $R^2$ or RMSE, as those were the hybrid techniques that we employed in the current study.

**Table 5.** Studies focused on research on heating load prediction.

| References | Article Title | Scope |
|---|---|---|
| Refs. [26,60] | Comprehensive preference learning and predicting heating load in residential buildings using machine learning techniques | Using traditional machine learning in predicting heating and cooling load |
| Refs. [57,107] | Proposing a novel predicting technique using M5Rules-PSO and M5Rules-GA model in estimating CL and HL in residential building system | Estimating cooling and heating load via a novel predictive technique using M5Rules |
| Ref. [108] | Predicting heating and cooling loads in residential buildings using two hybrid intelligent models | Hybrid intelligent models in predicting heating and cooling load |
| Ref. [109] | Optimal modification of HVAC system performances in energy-efficient buildings using the integration of metaheuristic optimization and neural computing | Using neural networks and metaheuristic optimization in modifying HVAC systems |
| Ref. [56] | Employing ABC and PSO techniques for optimizing a neural network in prediction of HL and CL of residential buildings | Using neural network algorithms in predicting cooling and heating load in residential green buildings |
| Ref. [39] | A teaching-learning based optimization Neural Processor for Predicting HL in Residential Buildings | Predicting heating load using a novel neural network algorithm of TLBO |

## 5. Conclusions

This study evaluates the MVO, SOSA, and VSA metaheuristic algorithms for analyzing and determining the HL. These methods served as the optimizer for a common neural predictive network simulation. The models predicted the HL based on a total of 768 design scenarios of the heating load. The following conclusions can be drawn from this work:

According to the sensitivity analysis, the MVA-MLP, SOSA-MLP, and VSA-MLP ensembles achieved optimal complexity at corresponding swarm sizes of 300, 500, and 250, respectively. The optimal MVO design required more calculation time than alternative MLP optimization algorithms. In terms of precision (MAEs of 0.08499, 0.19662, and 0.088861), all three ensembles profoundly understood the link between the HL and essential factors. During the testing phase, the measured value for the $R^2$ was 0.978, 0.895, and 0.977 demonstrating that the developed models were successful and had minimal prediction error. The most powerful model was the MVO-MLP, followed by the VSA-MLP and the SOSA-MLP. The MVO-MLP methodology was presented for use in real-world situations, but potential ideas for future projects were also presented in light of the shortcomings of the research, such as data enhancement and future selection, optimizing building characteristics using the model, and comparing the model to improved time-saving methods.

**Author Contributions:** F.N.: methodology, software, data curation. N.T.: writing—original draft preparation, investigation, validation. M.A.S.: conceptualization, methodology. A.G.: writing—original draft preparation, resources, final draft preparation. M.L.N.: supervision, project administration, funding acquisition. All authors have read and agreed to the published version of the manuscript.

**Funding:** This research received no external funding.

**Data Availability Statement:** The data used in this study is freely available on http://archive.ics.uci.edu/ml/datasets/Energy+efficiency (accessed on 15 July 2022).

**Conflicts of Interest:** The authors declare no conflict of interest.

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
