# Peer review of "Estimating Heating Load in Residential Buildings Using Multi-Verse Optimizer, Self-Organizing Self-Adaptive, and Vortex Search Neural-Evolutionary Techniques"

_buildings, doi:10.3390/buildings12091328_

Round 1
Reviewer 1 Report
This paper evaluates the accuracy of three models - MVO-MLP, SOSA-MLP, and VSA-MLP- for the heating load of Residential building. The paper is written in difficult way to understand, and so it needs to be rewritten and evaluated again.
1. It is difficult to understand the overall structure and description of the paper. There are many errors, and thus rewriting is required with the help of a professional translator.
2. The purpose of the paper was not described in the introduction
3. Typo correction is required.
- Page 6: Figure1. … with heating load with heating load
“Four novel optimizers” -> You only introduced three (MVO, SOSA, and VSA).
- Page10 : Table1. MSE -> RMSE
- ....
4. There is a lot of inconsistent and unknown contents.
- I don't know what parameter x1~x8 in Figure 1 means. -> If you simply draw a heat map, you can see the correlation between the models. Why is the data expressed this way?
There are many other things that need to be correct.
Re-examination is required after the correction.
Author Response
|
Reviewer #1: This paper evaluates the accuracy of three models - MVO-MLP, SOSA-MLP, and VSA-MLP- for the heating load of Residential building. The paper is written in difficult way to understand, and so it needs to be rewritten and evaluated again. Question: It isn't easy to understand the overall structure and description of the paper. There are many errors, and thus rewriting is required with the help of a professional translator. Answer: Thank you for your valuable comment. We have revised the manuscript thoroughly, and most sections are rewritten, and new statements are implemented to provide a better-revised version. After all comments have been considered, the manuscript was thoroughly edited by a professional English proofreading company. We believe that the revised manuscript is in a better form compared to its previous version. Question: The purpose of the paper was not described in the introduction. Answer: Thank you for the comment. In the revised manuscript, the last paragraph in the introduction section describes the purpose of this article. We have also added some other sentences to explain the purpose of the work more as follows. Environmentally and economically, finding a realistic model for thermal load modeling could be advantageous. The main goal of this article is to forecast the heating and cooling load via metaheuristic algorithms and check whether these algorithms can predict the heating and cooling load precisely. Metaheuristic optimizers, such as the multi-Verse Optimizer (MVO), Self-Organizing and Self-Adaptive (SOSA), and vortex search algorithm (VSA), are being evaluated to discover if they can aid in estimating the HL. Also, these three methods are compared with each other, and the best one is presented at the end of the task. Question: Typo correction is required. Answer: Thank you for your valuable comments. The typo errors have been corrected. a professional version of the Grammarly software was used to detect and correct typos. Moreover, as stated earlier, the revised manuscript was further edited by a professional English proofreader. Question: There is a lot of inconsistent and unknown contents. - I don't know what parameter x1~x8 in Figure 1 means. -> If you simply draw a heat map, you can see the correlation between the models. Why is the data expressed this way? Answer: Thank you for this valuable comment. The caption of Figure 1 has been updated, and all parameters including X1-X8 have been clearly defined. Clear definitions and explanations haven been added elsewhere throughout the text. |
Reviewer 2 Report
This manuscript contains the evaluation of the heating load (HL) of the heating, ventilation and air conditioning (HVAC) system of a residential building using a metaheuristic algorithm.
The authors used Multi-Layer Perceptron (MLP) neural networks with Multi-Verse Optimizer (MVO), Self-Organizing and Self-Adaptive (SOSA) and Vortex Search (VSA) algorithms.
In terms of performance, the MVO-MLP model outperformed the VSA-MLP and SOSA-MLP models.
It was interesting and useful. However, it was difficult for the reviewer to accept the generality of the results because the mechanism by which the differences between models were not clearly identified. The comments on the manuscript are as follows.
1. A careful analysis of why MVO is superior to VSA in the prediction of HL is necessary.
2. Does the result change depending on the number of hidden layers?
3. It is necessary to check whether all important factors affecting HL are considered.
4. Would the same conclusion be reached when predicting variables other than HL?
5. It seems to be necessary to compare other cases.
6. Performance was evaluated with R2, but the importance of HL prediction may not be uniform in all sections.
Author Response
|
Reviewer #2: This manuscript contains the evaluation of the heating load (HL) of the heating, ventilation and air conditioning (HVAC) system of a residential building using a metaheuristic algorithm. The authors used Multi-Layer Perceptron (MLP) neural networks with Multi-Verse Optimizer (MVO), Self-Organizing and Self-Adaptive (SOSA) and Vortex Search (VSA) algorithms. In terms of performance, the MVO-MLP model outperformed the VSA-MLP and SOSA-MLP models. It was interesting and useful. However, it was difficult for the reviewer to accept the generality of the results because the mechanism by which the differences between models were not clearly identified. The comments on the manuscript are as follows. Question: A careful analysis of why MVO is superior to VSA in the prediction of HL is necessary. Answer: Thank you for your valuable comment. In this article, the value of R2 for three methods, namely MVO, SOSA, and VSA is (0.977 and 0.978), (0.885 and 0.895), and (0.974 and 0.975) for testing and training phases, respectively. Also, in the case of RMSE, MVO, SOSA, and VSA have the value of (0.117 and 0.110), (0.255 and 0.239), and (0.124 and 0.112) in the training and testing phases, respectively. These results show that the highest value of R2 and the lowest value of RMSE are related to the MVO technique, indicating the best performance of MVO-MLP. According to R2 and RMSE values, the second technique for predicting HL and CL is VSA-MLP, and the last is SOSA-MLP. Question: Does the result change depending on the number of hidden layers? Answer: The number of hidden layers is one main criterion impacting the result's accuracy. That is the first item that needs to be optimized. So the answer is yes, changing the number of hidden layers during the calculation process can impact the results directly. However, once we have optimized it during the neural network development and training process, we don’t need to change it because we have already found the best-proposed number of hidden layers, which gives us the most accurate neural network predictive network for the next phase calculations. Question: It is necessary to check whether all important factors affecting HL are considered. Answer: Thank you. Yes, we have double-checked the calculations and tried to optimize all influential factors on the heating load calculations. I believe apart from the database we took from known resources, we have considered all the essential factors during the calculation process. Question: Would the same conclusion be reached when predicting variables other than HL? Answer: Thank you. Typically, each optimization technique has different variables, and we can provide a parametric study to optimize each item. But the main factor that optimizes the output the most is the swarm size, which is why researchers called these techniques intelligent swarm solutions. Therefore, we focused on optimizing the critical factor incorporated into the calculations the accuracies, such as the number of hidden layers in the neural network and the swarm size in the hybrid techniques calculations, instead of optimizing all factors. Question: It seems to be necessary to compare other cases. Answer: Thank you for this comment. A new table has been added to the discussion section and previous research has been added. Question: Performance was evaluated with R2, but the importance of HL prediction may not be uniform in all sections. Answer: Thank you. The results of accuracies are compared based on several statistical indices such as RMSE, MAE, and R2 in different parts. We used the same process for different groups, so the comparison had a logical context once all the calculations were performed for different cases. |
Round 2
Reviewer 1 Report
Thank you for editing paper.